# Synthesis, Characterization, and Biological Evaluation of Some Novel Pyrazolo[5,1-*b*]thiazole Derivatives as Potential Antimicrobial and Anticancer Agents

**DOI:** 10.3390/molecules26175383

**Published:** 2021-09-04

**Authors:** Abdulrhman Alsayari, Abdullatif Bin Muhsinah, Yahya I. Asiri, Faiz A. Al-aizari, Nabila A. Kheder, Zainab M. Almarhoon, Hazem A. Ghabbour, Yahia N. Mabkhot

**Affiliations:** 1Department of Pharmacognosy, College of Pharmacy, King Khalid University, Abha 61441, Saudi Arabia; alsayari@kku.edu.sa (A.A.); ajmohsnah@kku.edu.sa (A.B.M.); 2Department of Pharmacology, College of Pharmacy, King Khalid University, Abha 61441, Saudi Arabia; yialmuawad@kku.edu.sa; 3Department of Chemistry, College of Science, King Saud University, P.O. Box 2455, Riyadh 11451, Saudi Arabia; faizalaizari@yahoo.com (F.A.A.-a.); zalmarhoon@ksu.edu.sa (Z.M.A.); 4Department of Chemistry, Faculty of Science, Al-Baydha University, Albaydah 38018, Yemen; 5Department of Chemistry, Faculty of Science, Cairo University, Giza 12613, Egypt; nabila.abdelshafy@gmail.com; 6Department of Medicinal Chemistry, Faculty of Pharmacy, University of Mansoura, Mansoura 35516, Egypt; ghabbourh@yahoo.com; 7Department of Pharmaceutical Chemistry, College of Pharmacy, King Khalid University, Abha 61441, Saudi Arabia

**Keywords:** pyrazolo[5,1-*b*]thiazole, X-ray crystallography, antibacterial activity, antifungal activity, anticancer activity

## Abstract

The pharmacological activities of thiazole and pyrazole moieties as antimicrobial and anticancer agents have been thoroughly described in many literature reviews. In this study, a convenient synthesis of novel pyrazolo[5,1-*b*]thiazole-based heterocycles was carried out. The synthesized compounds were characterized by IR, ^1^H and ^13^C NMR spectroscopy and mass spectrometry. Some selected examples were screened and evaluated for their antimicrobial and anticancer activities and showed promising results. These products could serve as leading compounds in the future design of new drug molecules.

## 1. Introduction

Antibiotics saved millions of lives during the twentieth century by eliminating the deadly threat of infection. In recent years, the overuse of antimicrobial agents has played a significant role in creating more resistant strains of bacteria [1], thus causing an increase in morbidity and mortality [2]. Therefore, safer, cheaper, and more effective antimicrobial agents with a new mode of action are needed [3]. Although cancer is considered the second leading cause of death, taking the lives of 9.6 million people every year [4,5], many cancers are curable if detected early and treated promptly [6]. Chemotherapy is a treatment that uses medications to destroy cancer cells. It typically works by preventing cancer cells from developing, dividing, or proliferating. However, chemotherapy has several disadvantages, one of which is a lack of selectivity leading to extreme side effects and minimal efficacy. Another is the emergence of drug resistance [7]. Therefore, there is an urgent need to design and synthesize potent and highly selective anticancer molecules that offer little-to-zero toxicity to normal cells [8]. Thiazole derivatives demonstrate many pharmacological activities [9,10,11,12,13,14,15,16,17]. The thiazole ring can be traced in several well-established drugs such as the non-steroidal anti-inflammatory drug meloxicam, the anti-ulcer drug famotidine, the antibacterial sulfathiazole, the antiviral ritonavir, the antiparasitic thiabendazole, and many anticancer medicines including dasatinib, dabrafenib, and epothilones. Figure 1 shows some of the most effective drugs containing a thiazole ring [18]. 

Pyrazole derivatives have been reported as antimicrobial [19], analgesic [20], anti-inflammatory [21], and anticancer agents [22]. Additionally, many pharmaceutical drugs contain the pyrazole moiety, such as the antidepressant fezolamine and the anti-inflammatories celecoxib, mepirizole, and lonazolac. Moreover, the pyrazole derivative pyrazofurin has been reported to have antiviral [23,24] and anticancer activities [24,25]. Figure 2 depicts some of the most potent drugs containing a pyrazole ring.

Many literature reviews suggest that the pharmacophore hybrids may have enhanced efficacy, fewer drug–drug interactions, and less potential to induce drug resistance [26]. In light of the significance of pyrazoles and thiazoles, numerous studies have been conducted on the synthesis and biological evaluations of new hybrid pharmacophores containing pyrazole and thiazole moieties [27,28,29,30]. 

Figure 3 presents three examples of pharmacologically active pyrazolo[5,1-*b*]thiazole derivatives: pyrazolo[5,1-*b*]thiazole derivative (A) (a protein kinase inhibitor for treating cancer and other diseases) [31], pyrazolothiazole (B) (a potent corticotropin-releasing factor 1(CRF1) receptor antagonists) [32], and pyrazolo[5,1-*b*]thiazole derivative (C) (possessing a strong suppressant function against the H37Ra strain) [33] (Figure 3).

Hydrazides are an important class of biologically active compounds [34,35,36,37,38]. Hydrazides and their condensation products have been reported to possess a wide range of pharmacological and biological activities, including antibacterial [34], tuberculostatic [35], HIV inhibitory [36], pesticidal [37], and antifungal [38] activities. Some of them are used as monoamine oxidase (MAO) inhibitors and serotonin antagonists in psychopharmacology [39]. Furthermore, isonicotinoyl hydrazide (isoniazid) is an excellent antituberculosis drug [40,41,42]. A variety of methods have been used to form hydrazides [43]. The hydrazinolysis of carboxylic acid esters in alcohol solutions is a convenient method for preparing carbohydrazides [44]. In light of this, and as part of our ongoing research on pharmacologically potent molecules [45,46,47,48,49,50], new hydrazide–hydrazones attached to pyrazolothiazole were synthesized and evaluated for their antimicrobial and anticancer activities. 

## 2. Results

### 2.1. Chemistry

Hydrazide (**2**) was synthesized by treating diethyl 3,6-dimethylpyrazolo[5,1-*b*]thiazole-2,7-dicarboxylate (**1**) [51] with hydrazine hydrate (Scheme 1). The molecular structure was confirmed using IR, MS, and NMR analyses. Its IR spectrum showed the absence of C=O in the ester group, and the presence of absorption bands due to C=O in the amide and NHNH_2_ functions (see Experimental section). Another perfect confirmation of the structure formation obtained from NMR (^1^H and ^13^C) revealed the absence of any signals due to ethoxy protons and carbons. Additionally, the mass spectrum demonstrated the molecular ion peak at the expected m/z value of 268 (41%). Compound **2** was reacted with the appropriate aromatic aldehyde to afford the corresponding hydrazones **3a**,**b** (Scheme 1). Their ^1^H NMR spectra revealed the absence of amino signals, which also confirms the presence of a signal at (6.78–6.81 ppm) for N=CH (imine group) in the hydrazone compounds.

Hydrazide **2** was reacted with phenyl isothiocyanate in ethanol and in the presence of a catalytic amount of triethylamine to afford *O*-ethyl *N*-phenylcarbamothioate (**4**) [52], rather than the expected product **5** (Scheme 2). The structure was confirmed using spectral and X-ray analysis (Figure 4). CCDC 2075096 contains the supplementary crystallographic data for this paper. These data can be obtained free of charge from the Cambridge Crystallographic Data Centre via www.ccdc.cam.ac.uk/data_request/cif. Additional information relating to compound **4** is provided in Table 1.

The ring closure reaction of acid hydrazide **2** with carbon disulfide in ethanolic KOH afforded the target compound **6** (Scheme 3). It was observed that 1,3,4-oxadiazole-2-thione derivatives exist in the thione form in solution, rather than in the thiol form [53,54]. Additionally, the thione tautomer is more stable than the thiol in the solution [54,55]. The equilibrium is even more favored towards the thione as it is better solvated than the thiol form [54]. In the ^1^H NMR spectrum of compound **6** (Scheme 3), a signal at δ 12.9 of the NH proton was recorded.

The reaction of hydrazide **2** with ethyl cyanoacetate in absolute ethanol resulted in compound **7** as the sole product (Scheme 4). Its IR spectrum showed the absence of any absorption band due to the cyano group and the presence of the stretching bands at 3164, 1726, and 1651 cm^−1^, corresponding to NH and two C=O groups, respectively. Its mass spectrum demonstrated the molecular ion peak at an m/z value of 402. On the other hand, hydrazide **2** was converted to acyl azide **8** in the presence of sodium nitrite and acetic acid (Scheme 4). The reaction between compound **8** and ethyl acetoacetate or ethyl cyanoacetate afforded the target compounds **9** and **10**, respectively (Scheme 4). The structures of compounds **8–10** were confirmed through analytical data and spectral analysis (See Experiment section). The suggested mechanism for the selective synthesis of compounds **9** and **10** via the reaction of hydrazide **8**, ethyl acetoacetate, or ethyl cyanoacetate in the presence of sodium ethoxide is outlined in Scheme 5 [56]. The reaction was assumed to proceed through a concerted [3+2]-cycloaddition reaction. The non-isolable intermediate **11** was further transformed into stable 1,2,3-triazole derivative **9** through rapid elimination of two water molecules induced by sodium ethoxide.

### 2.2. Biological Activity Evaluation

#### 2.2.1. Anticancer Screening of the Synthesized Compounds

The in vitro anti-tumor activity of the synthesized compounds was assessed against two human cancer cell lines: human hepatocellular carcinoma cell line (HepG-2) and colon carcinoma cell line (HCT-116), using the MTT assay [57]. Their activity was compared to the reference drug Doxorubicin. In addition, calculations of the tested compounds’ concentrations needed to inhibit 50% of the cancerous cell population (IC_50_) were implemented. These are presented in Table 2 and Table 3.

Of all the tested compounds, 1,3,4-oxadiazole derivative **6** exhibited the highest activity against the two tested cell lines; HepG-2 and HCT-116, with an IC_50_ = 6.9 and 13.6 µg/mL, respectively. Having azide moiety, pyrazolothiazole derivative **8** revealed high activity against HepG-2 and HCT-116, with an IC_50_ = 12.6 and 28.9 µg/mL, respectively.

These results support previously published results indicating that compound structures containing an 1,3,4-oxadiazole ring [58] or azide moiety [59] have potent antitumor activities.

#### 2.2.2. The in Vitro Antimicrobial Assessments

Assessments of the antimicrobial activities of the synthesized compounds were performed using the inhibition zone technique [60] against six species: two fungal species (*Aspergillus fumigatus* (RCMB 002008 (4) and *Candida albicans* (RCMB 05036)), two Gram-positive bacteria (*Staphylococcus aureus* (RCMB010010 and *Bacillus subtilis* (RCMB 010067)), and two Gram-negative bacteria (*Salmonella SP*. (RCMB 010043) and *Escherichia coli* (RCMB 010052)). The standard drugs used for comparison were Amphotericin B, Gentamicin, and Ampicillin. The inhibition zone diameter (IZD) was used as the criterion for the antimicrobial activity and all results are summarized in Table 4.

The results of Table 4 illustrate the following points:All the tested compounds except compound **7** showed excellent activity against *Aspergillus fumigatus*. Compounds **4** and **6** were especially effective.All tested compounds except compound **8** showed high antifungal activity against *Candida albicans*.Compounds **3b**, **7**, and **8** were found to be more active against *Staphylococcus aureus* than against *Bacillus subtilis*.The best antibacterial activity was observed for compounds **4** and **6**: their inhibitory effect appears to be equipotent to Gentamycin against *Salmonella SP* and *Escherichia coli*.

Many thiocarbamate derivatives such as tolnaftate, tolciclate, and piritetrade are commonly used as fungicidal agents, and this explains the highest antifungal activity of compound **4** [61,62,63,64,65].

## 3. Materials and Methods

### 3.1. Chemistry

#### 3.1.1. Materials and Equipment

See Appendix A.

#### 3.1.2. Synthesis of 3,6-Dimethylpyrazolo[5,1-b]thiazole-2,7-dicarbohydrazide (**2**)

A mixture of diethyl 3,6-dimethylpyrazolo[5,1-*b*]thiazole-2,7-dicarboxylate (**1**) [51] (1.48g, 5 mmol), and hydrazine hydrate (80%, 15 mmol) in ethanol (15 mL) were refluxed for 3 h. Excess ethanol was evaporated under reduced pressure and the solid product was filtered, dried, and recrystallized from ethanol/DMF to afford target compound **2** at 90% yield; mp: 210–211 ^°^C; IR (KBr) νmax 3338–3201(NH_2_+NH), 1717 (C=O) cm^−1^; ^1^H NMR (CDCl_3_): δ 2.04 (s, 3 H, CH_3_), 2.16 (s, 3 H, CH_3_), 3.31 (s, 4H, NH_2_), 11.58 (s, 2 H, 2NH); ^13^C NMR (CDCl_3_): δ 13.52 (CH_3_), 15.68 (CH_3_), 128.55, 129.71, 137.77, 137.86, 147.46, 186.80 (C=O); MS *m*/*z* (%) 268 (M^+^, 41%), 251 (100%). Anal. Calcd. for C_9_H_12_N_6_O_2_S (268.30): C, 40.29; H, 4.51; N, 31.32. Found: C, 40.33; H, 4.62; N, 31.25.

#### 3.1.3. Synthesis of Hydrazones **3a**,**b**

A mixture of hydrazide **2** (0.536g, 2 mmol) and appropriate aldehydes (4.2 mmol) in absolute ethanol/DMF (20mL) were refluxed for 5 h. The resulting precipitate was filtered off, washed, dried, and recrystallized by DMF/ethanol to afford the corresponding hydrazones **3a**,**b**.

**3a**: Yield (72%), mp. ˃ 300 °C; IR (KBr) νmax 3274 (NH), 1714 (C=O), 1609 (C=N) cm^−1^; 1H-NMR (CDCl_3_): δ 1.92 (s, 3H, CH_3_), 2.20 (s, 3H, CH_3_), 6.78 (s, 2H, 2CH), 7.23–7.85 (m, 10H, ArH), 10.78 (s, 1H, NH), 11.75 (s, 1H, NH); ^13^C-NMR: δ 11.02 (CH_3_), 14.0 (CH_3_), 111.12, 128.5, 129.8, 130.3, 133.1, 135.7, 136.0, 146.2, 151.4, 163.0, 167.2 (C=O). Anal. Calcd. for C_23_H_20_N_6_O_2_S (444.51): C, 62.15; H, 4.54; N, 18.91. Found: C, 62.22; H, 4.42; N, 18.77.

**3b**: Yield (80%), mp. 260 °C; IR (KBr) νmax 3515 (NH), 1686 (C=O), 1595 (C=N) cm^−1^; ^1^H-NMR (CDCl_3_): δ, 1.90 (s, 3H, CH_3_), 2.20 (s, 3H, CH_3_), 2.45 (s, 6H, 2CH_3_), 6.81 (s, 2H, 2CH), 7.28–7.84 (m, 8H, ArH), 11.25 (s, 2H, 2NH); ^13^C-NMR: δ 11.50 (CH_3_), 14.31 (CH_3_), 18.95 (2CH_3_), 111.30, 128.1, 129.7, 131.20, 133.1, 136.12, 136.7, 148, 151.0, 156.0, 163.0, 165.61 (C=O). Anal. Calcd. for C_25_H_24_N_6_O_2_S (472.56): C, 63.54; H, 5.12; N, 17.78. Found: C, 63.34; H, 5.22; N, 17.87.

#### 3.1.4. Synthesis of *O*-Ethyl *N*-phenylcarbamothioate (**4**)

A mixture of hydrazide (**2**) (0.268g, 1 mmol) and phenyl isothiocyanate (0.27 g, 0.24 mL, 2 mmol) in EtOH (10 mL), in the presence of a few drops of triethylamine as a catalyst, was heated under reflux for 5 h and then allowed to cool to room temperature. The precipitated solid was filtered off, dried, and recrystallized from methanol to afford compound **4** [52] at 20% yield; mp: 55–56 °C; IR (KBr) νmax 3212 (NH), 3039, 2982 (CH) cm^−1^; ^1^H NMR (CDCl_3_): δ 1.23 (t, 3 H, CH_3_), 4.65 (q, 2 H, CH_2_), 7.20–7.46 (m, 5H, ArH), 9.12 (s, 1 H, NH); ^13^C NMR (CDCl_3_): δ14.90, 68.50, 121.10, 121.10, 128.40, 129.57, 129.57, 137.85 (Ar-C), 188.50 (C=S). Anal. Calcd. for C_9_H_11_NOS (181.25): C, 59.64; H, 6.12; N, 7.73. Found: C, 59.75; H, 6.07; N, 7.88.

#### 3.1.5. Synthesis of Bis(1,3,4-oxadiazole) **6**

Hydrazide **2** (3.75 g, 14 mmol) was dissolved in absolute ethanol (50 mL). CS_2_ (2.7g, 2.1 mL, 35 mmol) was then added to the solution, followed by the addition of a KOH solution (1.6 g, 28 mmol) in water (20 mL). The reaction mixture was thoroughly stirred and refluxed for 3 h until the evolution of H_2_S ceased. After completion of the reaction, excess ethanol was removed under reduced pressure. The mixture was poured into a mixture of H_2_O/ice and acidified with concentrated HCl. The precipitated solid was filtered off and recrystallized from ethanol/DMF, resulting in thione **6**. Yield (40%); mp. 250 ^o^C; IR (KBr) νmax 3313 (NH), and 1116 (C=S) cm^−1^; ^1^H NMR (CDCl_3_): δ 1.31 (s, 3H, CH_3_), 2.48 (s, 3H, CH_3_), 12.90 (s, 2H, NH); ^13^C-NMR δ 11.50 (CH_3_), 14.01(CH_3_), 104.02, 107.30, 135.11, 142.90, 156.74, 162.21, 177.77 (C=S). Anal. Calcd. for C_11_H_8_N_6_O_2_S_3_ (352.42): C, 37.49; H, 2.29; N, 23.85; Found: C, 37.38; H, 2.18; N, 23.99.

#### 3.1.6. Synthesis of Bis(pyrazole) Derivative **7**

A mixture of hydrazide **2** (1.34 g, 5 mmol) and ethyl cyanoacetate (2.26 mL, 20 mmol) in ethanol (10 mL) was heated under reflux for 5 h. The precipitated solid product was filtered and recrystallized from ethanol, resulting in **7** at 55% yield; mp. 250–251 °C; IR (KBr) νmax 3164 (NH), 2982 (CH aliphatic), 1726, 1651 (2C=O), 1560 (C=N) cm^−1^; MS *m*/*z* (%) 402 (M^+^, 4%), 400 (52%), 399 (35%), 45 (100%). Anal. Calcd. for C_15_H_14_N_8_O_4_S (402.39): Calc.: C, 44.77; H, 3.51; N, 27.85. Found: C, 44.65; H, 3.42; N, 27.93.

#### 3.1.7. Synthesis 3,6-Dimethylpyrazolo[5,1-b]thiazole-2,7-dicarbonyl Azide (**8**)

To a suspension of hydrazide **2** (2.68 g, 10 mmol) in 30 mL of H_2_O, 3.5 g (50 mmol) of sodium nitrite was added. The mixture was then cooled in ice and treated portion-wise with 3 mL (50 mmol) of acetic acid. After stirring at room temperature for 3 h, the resulting precipitate **8** was filtered, washed with H_2_O, and dried: yield 90%; mp 120–121 °C; IR (KBr) νmax 2982 (CH), 2167 (N=N), 1724 (C=O), 1686 (C=O) cm^−1^; MS *m*/*z* (%) 290 (M^+^, 14%), 40 (100%). Anal. Calcd for C_9_H_6_N_8_O_2_S (290.26): C, 37.24; H, 2.08; N, 38.60. Found: C, 37.35; H, 2.16; N, 38.49.

#### 3.1.8. Synthesis of Bis(1,2,3-triazole) Derivatives **9**,**10**

Compound **8** (0.290 g, 1 mmol) was added to a stirred solution of sodium metal (0.10 g) in ethanol (20 mL), and the mixture was left to stir at room temperature for 20 min. Either ethyl acetoacetate or ethyl cyanoacetate (2 mmol) was added while stirring. The reaction mixture was then left to stir for a further 24 h. The formed solid product was filtered off, washed with water, dried, and recrystallized from EtOH to afford the corresponding bis(1,2,3-triazole) derivatives **9** and **10**, respectively.

**9.** Yield (72%), mp. 300 °C; IR (KBr) νmax 1694 (2C=O), 1594 (C=N) cm^−1^; ^1^H-NMR (CDCl_3_): δ 1.31 (s, 6H, 2CH_3_), 2.26 (s, 6H, 2CH_3_), 2.38 (s, 6H, 2CH_3_), 4.32 (q, 4H, 2CH_2_); ^13^C-NMR: δ 13.40, 14.3, 15.10, 18.10, 60.75, 111.10, 130.0, 134.0, 137.20, 138.0, 145.00, 151.15, 164.93, 169.34 (C=O). Anal. Calcd. for C_21_H_22_N_8_O_6_S (514.51): C, 49.02; H, 4.31; N, 21.78. Found: C, 49.13; H, 4.37; N, 21.88.

**10.** Yield (75%), mp. 210–211 °C; IR (KBr) νmax 1697 (2C=O), 1597 (C=N) cm^−1^; ^1^H-NMR (CDCl_3_): δ 1.19 (s, 6H, 2CH_3_), 2.46 (s, 3H, CH_3_), 3.30 (s, 3H, CH_3_), 4.28 (q, 4H, 2CH_2_), 7.71 (s, 4H, NH_2_); ^13^C-NMR: δ 12.40, 15.10, 19.44, 60.23, 61.75, 111.30, 131.20, 133.89, 137.20, 138.20, 146.00, 151.02, 165.93, 169.20 (C=O). Anal. Calcd. for C_19_H_20_N_10_O_6_S (516.49): C, 44.18; H, 3.90; N, 27.12. Found: C, 44.22; H, 3.84; N, 27.23.

### 3.2. Biological Tests

#### 3.2.1. Evaluation of Antitumor Activity

The MTT assay was used to investigate the in vitro antitumor activity of the synthesized compounds against two human cancer cell lines: human hepatocellular carcinoma cell line (HepG-2) and colon carcinoma cell line (HCT-116) [57].

#### 3.2.2. Antimicrobial Evaluation

The inhibition zone technique [60] was used to evaluate the antimicrobial activity of the synthesized compounds against six pathogens. Amphotericin B, Gentamicin, and Ampicillin were the standard medications utilized for comparison. The antimicrobial activity was measured using the inhibition zone diameter (IZD).

## 4. Conclusions

New pyrazolo[5,1-*b*]thiazole derivatives, synthesized using simple synthetic methods, can be used as leading compounds in the development of future, novel drug molecules.

## Data Availability

The data presented in this study are available on request from the corresponding author.

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
