# Peer review of "Synthesis, Characterization, and Biological Evaluation of Some Novel Pyrazolo[5,1-b]thiazole Derivatives as Potential Antimicrobial and Anticancer Agents"

_molecules, 2021, doi:10.3390/molecules26175383_

Round 1

Reviewer 1 Report

In this form the manuscript does not correspond to the requirements of this journal. There are some novelty elements (synthesis of new organic compounds), but the attribution of the structure must be completed by Physico-chemical studies (eg solubility).

In terms of biological activity it is very briefly presented.     The solvent used to dissolve the compounds must be specified to see if it is compatible with the cell culture medium, what concentrations of compounds were used, etc.    I also think it is necessary to test the cytotoxicity of the compounds on normal cells and only then to perform antitumor activity studies.

Antibacterial activity: solvent used, concentrations of compounds used, more detailed explanations of how it works.

More detailed presentation for the studied compounds of the results and discussions of the presented biological activity (structure-biological activity relations, explanation of the superior activity of compound 4 compared to the others, etc.)            

Author Response

The authors have provided detailed responses to referee comments in the attachment. 

Reviewer 2 Report

Dear authors,

I would like to congratulate you for the interesting topic of your paper and the hard work regarding the synthesis and biological evaluation of these novel heterocyclic compounds. Chemically speaking, the nature of the compounds and their synthesis cannot be characterised as original, nevertheless these kinds of scaffolds are indeed of high importance in medicinal chemistry. There are some minor improvements that could be done, which I will try to summarize in the following points:

  1. In my opinion as a reader, the abstract should not be focused in the chemical processes of numbered compounds that cannot be seen in the results of search engines. eg." hydrazide 2 was prepared " etc. It should be focused in the scope of the research and related matters.
  2. In the title, X-ray analysis is mentioned in a way that implies that it was performed for all synthesized compounds. I am not sure if it should be removed, since as I saw X-ray analysis was performed for 1 only compound (if I understood correctly).
  3. More references are needed in th intro regarding antibacterials, antifungals and anticancer compounds.
  4. Line 78: "the most common method" is a strong phrase thats need to be supported more or to be rephrased.
  5. Line 115: regarding the higher sollubility of the thione, you could add some experimental or theoritical/prediction data to support your theory.
  6. Why compounds 9 and 10 were not tested with the MTT assay? Was there a special reason?
  7. Same for antimicrobial activity? Why was compound 9 excluded?

I hope you find those comments helpful and use the in order to improve the final version of this interesting paper.

Kind regards

Author Response

(The authors gave the same response as above.)

Round 2

Reviewer 1 Report

In the previous review there were observations regarding the improvement of the experimental part and the discussions. You say you will do this in future articles or future research. In my opinion, this article, in order to be complete and complex, must contain these experimental data and elaborate discussions.
